# Dynamics and mechanism of dimer dissociation of photoreceptor UVR8

Xiankun Li [1,2], Zheyun Liu[1], Haisheng Ren [3,4], Mainak Kundu[1], Frank W. Zhong[5], Lijuan Wang[1], Jiali Gao [3,6✉] & Dongping Zhong [1,2✉]

Photoreceptors are a class of light-sensing proteins with critical biological functions. UVR8 is the only identified UV photoreceptor in plants and its dimer dissociation upon UV sensing activates UV-protective processes. However, the dissociation mechanism is still poorly understood. Here, by integrating extensive mutations, ultrafast spectroscopy, and computational calculations, we find that the funneled excitation energy in the interfacial tryptophan (Trp) pyramid center drives a directional Trp-Trp charge separation in 80 ps and produces a critical transient Trp anion, enabling its ultrafast charge neutralization with a nearby positive arginine residue in 17 ps to destroy a key salt bridge. A domino effect is then triggered to unzip the strong interfacial interactions, which is facilitated through flooding the interface by channel and interfacial water molecules. These detailed dynamics reveal a unique molecular mechanism of UV-induced dimer monomerization.

[1] Department of Physics, Department of Chemistry and Biochemistry, Programs of Biophysics, Chemical Physics and Biochemistry, The Ohio State University, Columbus, OH 43210, USA. [2] Center for Ultrafast Science and Technology, School of Physics and Astronomy, School of Chemistry and Chemical Engineering, Shanghai Jiao Tong University, Shanghai 200240, China. [3] Department of Chemistry and Supercomputing Institute, University of Minnesota, Minneapolis, MN 55455, USA. [4] College of Chemical Engineering, Sichuan University, Chengdu 610065, China. [5] Cell and Molecular Biology Program, University of Chicago, Chicago, IL 60637, USA. [6] School of Chemical Biology and Biotechnology, Peking University Shenzhen Graduate School, Shenzhen 518055, China. ✉email: jiali@jialigao.org; zhong.28@osu.edu

Sunlight is essential to sessile plants both as an energy source to fuel photosynthesis for energy conversion[1–3] and as a biological signal to regulate growth and development in their life cycle[4–7]. For harmful ultraviolet (UV) radiation, plants perceive UV-B (280–315 nm) fraction by a photoreceptor UVR8, inducing gene regulation for prevention of UV-induced damages[8–10]. Recent X-ray structure of UVR8 photoreceptor shows a homodimer configuration (Fig. 1a)[11,12]. Uniquely, UVR8 does not have any external chromophores and uses natural amino-acid tryptophan (Trp or W) for light sensing, suggesting a distinct light-perception mechanism. The dimer interface is glued by interprotein interactions of salt bridges, hydrogen-bond networks, and hydrophobic clusters with confined water molecules (Fig. 1a, b). Upon photoreception, dimeric UVR8 dissociates into two monomers to interact with downstream proteins for subsequent signal transduction[8–15]. However, the UV-induced dissociation mechanism is still poorly understood, and several hypotheses are highly controversial[11,12,16–19]. Although all previous models suggest the breaking of critical cross-dimer salt-bridges, different mechanisms were proposed: disruption of cation-π interactions by tryptophan excitation[11]; electron transfer from excited tryptophan to nearby arginine[12]; formation of neutral W233/Arginine (Arg or R) diradicals via electron transfer and proton transfer[16]; W285$^{+\bullet}$/W233$^{-\bullet}$ charge-transfer state electrostatically drives proton transfer in the salt-bridging residues[18]; proton-coupled electron transfer (PCET) from W285 triplet state to R286[19]. Here, we integrate extensive site-directed mutations (Supplementary Tables 1 and 2), femtosecond (fs) spectroscopy and computational calculations, and finally reveal its molecular mechanism of UV-B perception for dimer dissociation.

## Results

### Phenylalanine scanning and critical fluorescence quenching of W285 and W233.

UVR8 contains fourteen Trp residues and besides one Trp in the random coil of the C-terminal it has three unique groups of Trp residues[20]: six distal buried Trp residues (W39, W92, W144, W196, W300, and W352) form a core ring in the middle of beta-sheets, three central interfacial Trp residues (W233, W285, and W337) condense into a so-called "Trp pyramid" or "Trp cluster" with another Trp (W94) from the opposing monomer (Fig. 1b), and the other three peripheral interfacial Trp residues (W198, W250, and W302) encircle the pyramid. To investigate the functional roles of these Trp residues, except the conserved distal W352, we have systematically mutated thirteen other Trp residues one at a time. Except W285F and W233F, all eleven single-Trp mutants exhibit similar fluorescence emission spectra (Fig. 1c) and decay dynamics (Supplementary Fig. 1) to the wild type (WT) and maintain the normal function of UV-induced dimer dissociation (Supplementary Fig. 1), indicating that only W285 and W233 are critical to UVR8 monomerization. Mutation of W285 or W233 leads to higher fluorescence quantum yield ($Q_F$) than WT (Fig. 1c inset), suggesting a quenching reaction between W285 and W233. Our previous studies have revealed a unique light-harvesting Trp network, which funnels all excitation energy to W285 and W233

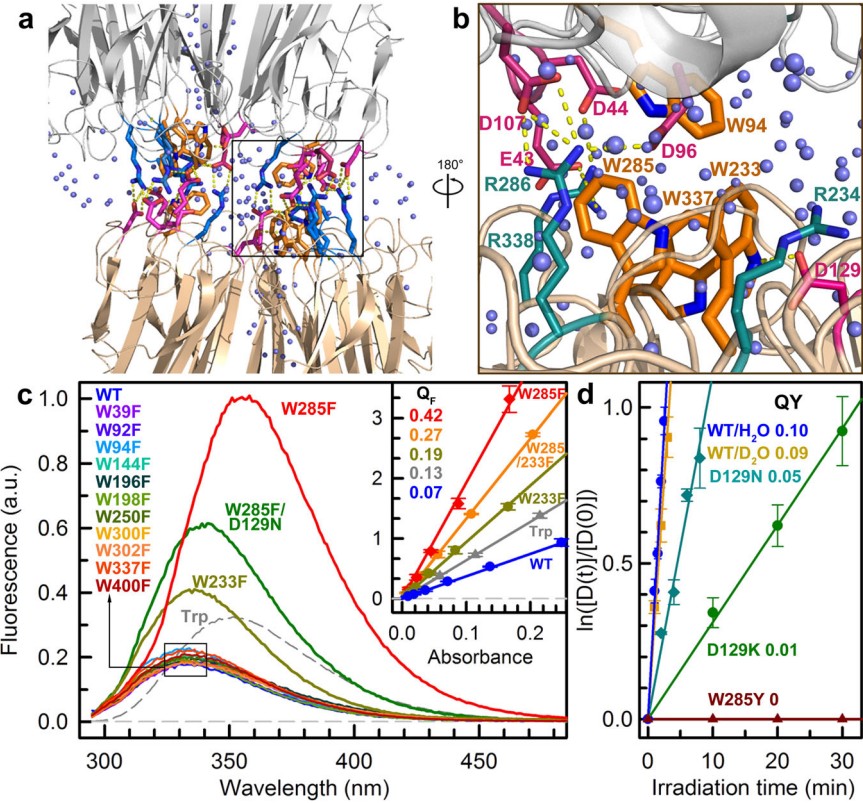

**Fig. 1 UVR8 structure, emission spectra and quantum yields. a** Overall structure of UVR8 homodimer with two tryptophan pyramid centers (orange), interfacial acidic residues (pink), interfacial basic residues (blue) and water molecules (light blue) highlighted. **b** Local structure around the tryptophan pyramid center. Two critical salt bridges, R286-D96/D107 and R338-E43/D44, are near W285. D129 and R234 are close to W233. D129 is hydrogen-bonded to W233 indole ring. **c** Steady-state emission spectra of WT, single Trp mutants and W285F/D129N are shown in various colored lines. The emission of free Trp in buffer was also showed in a dashed line for comparison. The inset shows fluorescence quantum yields ($Q_F$) of WT, free Trp and selected mutants with the standard deviation of five measurements. **d** Dissociation kinetics and total quantum yields (QY) of various UVR8 samples with the standard deviation of five (wild type) and three (all others) measurements.

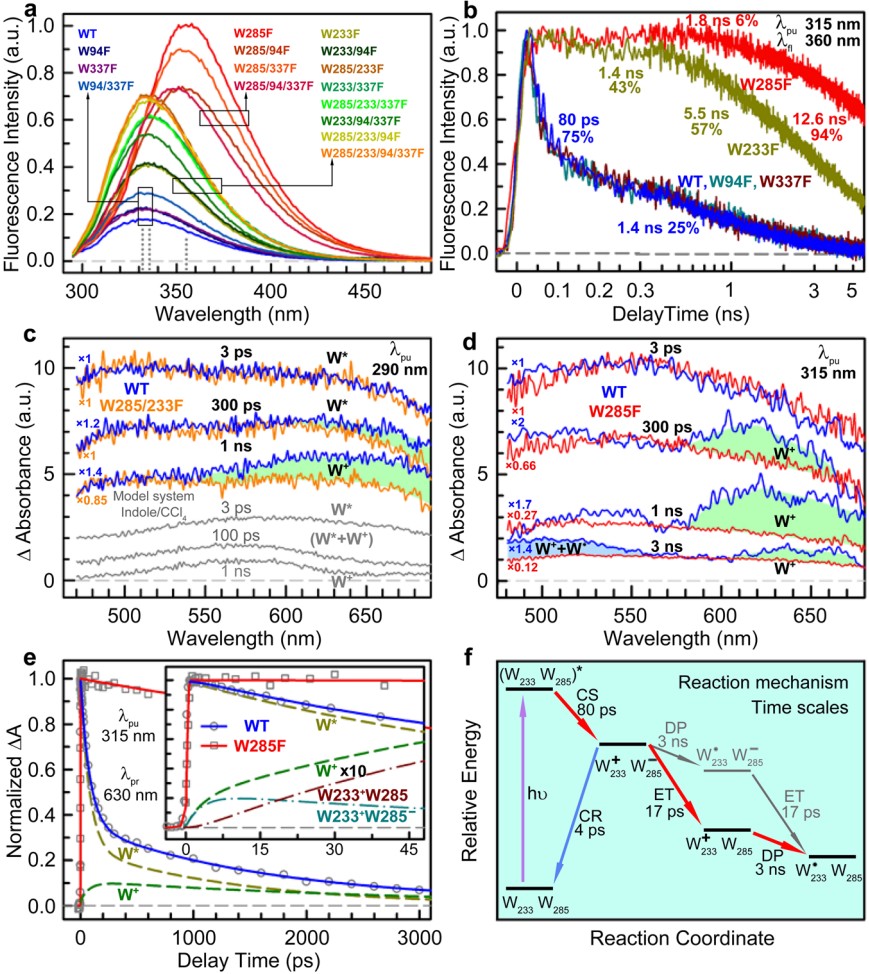

**Fig. 2 Mutation scan, charge separation, and subsequent reactions. a** Steady-state emission spectra ($\lambda_{pu} = 290$ nm) of WT and all 15 mutants are shown in different colors. Three groups of mutants: with both W285 and W233, without W233, and without W285. Their emission peaks are ~332 nm, ~336 nm, and ~354 nm, respectively (dotted lines). **b** Fluorescence decay transients of WT and four mutants at 315-nm excitation. The fitting results are also shown. Note the critical 80-ps quenching component only in the presence of both W285 and W233. The delay time is shown on a linear scale before 300 ps and a logarithmic scale thereafter. **c** Broadband transient absorption spectra of WT (blue lines), W285/233F (orange lines), and indole/CCl$_4$ model system (gray lines) at various delay times at 290-nm excitation. For shape comparison, absolute intensities of the spectra were adjusted by multiplying by different factors, as labeled near the left side of each spectrum. The spectral difference due to Trp cation radicals (W$^{+\bullet}$) was shaded by light green color. **d** Broadband transient absorption spectra of WT (blue lines), W285F (red lines) at various delay times at 315-nm excitation. For shape comparison, absolute intensities of the spectra were adjusted by multiplying by certain factors, as labeled near the left side of each spectrum. The spectral difference due to Trp cation radicals (W$^{+\bullet}$) and by Trp neutral radicals was shaded in light green and light blue colors, respectively. **e** The transient dynamics of WT and W285F probed at 630 nm upon 315-nm excitation. For WT, the transient was deconvoluted into two components of Trp excited state W* (dark yellow dashed line) and Trp cation radical W$^{+\bullet}$ (green dashed line). The symbols are experimental data and the lines are best fit. The first 48-ps data are shown in the inset. The W$^{+\bullet}$ signal and its components of W233$^{+\bullet}$W285 (dark red dash-dot line) and W233$^{+\bullet}$W285$^{-\bullet}$ (cyan dash-dot line) were multiplied by 10 for clarity. Note the initial ~3.2-ps rise in WT. **f** Reaction mechanism and time scales. CS-charge separation between W285 and W233; ET-electron transfer from W285$^{-\bullet}$ to positive R286$^+$/R338$^+$; CR-charge recombination of W285$^{-\bullet}$ and W233$^{+\bullet}$; DP-deprotonation of W233$^{+\bullet}$ to D129. Time scales for each reaction step are shown. The charge recombination (CR) step is a futile branching.

in the pyramid center from other distal and peripheral Trp residues[20,21]. The WT dissociation quantum yield (QY) is determined to be 0.1 (Fig. 1d), consistent with the value by the other method[22]. In deuterium oxide (D$_2$O), the QY value is similar, suggesting that the proton transfer does not involve in critical elementary reactions during dimer dissociation. The mutation of D129, hydrogen-bonded to W233, destabilizes excited W233 (Fig. 1c) and results in lower QY (Fig. 1d).

To elucidate the roles of W285 and W233 in dissociation, we further made all combinations of mutations in the pyramid (Fig. 2a). Clearly, UVR8 shows the UV-induced dissociation function only when both W285 and W233 are present (WT, W94F, W337F, and W94/337F); all other mutants without W285

or W233 or both lose dissociation function. Without W285, all excitation energy funnels into W233 with an emission peak at 355 nm and a longest lifetime of 12.6 ns observed so far in protein (Supplementary Fig. 2). With both W285 and W233, all excitation energy is delocalized in the cluster to form an exciton as observed by our circular dichroism and also other studies[12,23]. With selective 315-nm excitation of the pyramid center Trp only[20], the fluorescence dynamics of WT and four single Trp mutants are shown in Fig. 2b. Clearly, both W94F and W337F show the same fluorescence decay dynamics as WT, with an 80-ps component (75%) of the quenching reaction[21,24], and a 1.4-ns effective lifetime component (25%) resulting from structural fluctuations of the Trp pyramid. Significantly, the mutation of

either W285 or W233 causes the 80-ps reaction to vanish, confirming that a critical photochemical reaction occurs between W285 and W233. The transient of W285F mainly reflects the W233 fluorescence lifetime decay with a dominant 12.6-ns component (94%) while the transient of W233F shows the other three Trp fluorescence dynamics with shorter lifetimes without the energy sink[20] of W233.

**Trp-Trp charge separation and tryptophan anion formation (W285$^{-\bullet}$).** To determine the quenching mechanism, we propose an exciton-induced charge separation (80 ps) between W285 and W233, and thus need to capture the reaction intermediates. There has been no experimental report in literature of Trp-Trp charge separation upon UV excitation but the Trp cation (W$^{+\bullet}$) absorption is known[25–27]. We first measured the broadband transient-absorption spectra for WT and W285/233F (as a control) with 290-nm excitation (Fig. 2c). We used a well-established model system of indole/CCl$_4$ and observed cationic Trp (W$^{+\bullet}$) formation with a peak at 580 nm after the same excitation[25]. At 3 ps, the absorption spectra of WT and W285/233F are nearly similar, showing the featureless absorption of excited-state Trp (W*). At later delay times of 300 ps and 1 ns, we observed a W$^{+\bullet}$ absorption band around 620 nm in WT, red-shifted by 40 nm in the pyramid cluster at the protein interface. This characteristic band was not observed in the control sample of W285/233F or WT monomer (Supplementary Fig. 3). Here, the W$^{+\bullet}$ signal is not prominent at 290-nm excitation due to the dominant contributions from W* (87%), as demonstrated by our numerical simulations (Supplementary Fig. 4). To minimize the contributions of W* signals from distal and peripheral Trp residues, we selectively excited Trp in the pyramid center with 315-nm excitation and measured the broadband transient-absorption spectra at various delay times (Fig. 2d and Supplementary Fig. 5). Similarly, WT and W285F have similar W* absorption at 3 ps. But, unlike 290-nm excitation, significant W$^{+\bullet}$ absorption signals were observed at 300 ps, 1 ns and 3 ns. Notably, no W$^{+\bullet}$ was observed in the mutant of W285F, implying that the ET reaction must occur between W285 and W233 in 80 ps. Recent experiments suggest formation of W285 triplet state via intersystem crossing, followed by proton-coupled electron transfer (PCET), with salt-bridging R286 as the electron acceptor and D96 as the proton acceptor, neutralizing the charges on both residues[19]. The fast W* decay and W$^{+\bullet}$ accumulation observed here indicate direct conversion from W singlet excited state to W$^{+\bullet}$ in 80 ps and thus rules out this PCET mechanism[19]. We argue that the observed Trp triplet state signal is mainly from distal and peripheral Trp, since the triplet absorption was also observed in the mutant of W233/285/337F.[19]

W285 is flanked by two neighboring positively charged arginine residues of R286$^+$ and R338$^+$, and thus is more likely to be an electron acceptor; W233 could be the electron donor since W233$^{+\bullet}$ is stabilized through a hydrogen bond with the negatively charged D129. Interestingly, at 3 ns, the deprotonated Trp neutral radical was observed at ~500 nm,[27,28] indicating that following initial charge separation, deprotonation of W233$^{+\bullet}$ to D129 occurs at the nanosecond timescale. We performed QM/MM calculations and predicted the electron transfer from excited W233 to W285 in 71 ps (Supplementary Table 3), which is extremely close to our observed value of 80 ps. The charge-transfer (CT) direction here is opposite to a previous computational study that suggests W233$^{-\bullet}$/W285$^{+\bullet}$ CT state causes local electrostatic perturbations, driving proton migration from hydrogen-bonded Arg to Asp.[18] To further examine this hypothesis, we engineered permanent charges in two mutants of W285K/W233D and W285D/W233K and found both mutants

are dimeric (Supplementary Fig. 6) without monomerization under UV irradiation, suggesting that the change of electrostatic potential alone cannot drive UVR8 monomerization and thus subsequent electron transfer is needed to neutralize nearby Arg. Based on these findings, we conclude that W285$^{-\bullet}$/W233$^{+\bullet}$ is formed in the 80 ps via electron transfer, and subsequently the anionic W285$^{-\bullet}$ must continue to react with the neighboring positive R286$^+$ or R338$^+$ to break the critical salt bridge(s) and trigger dissociation[11].

**Subsequent arginine neutralization, charge recombination, and quantum efficiency.** The formed W285$^{-\bullet}$ anion can proceed along two reaction pathways: the electron either jumps back to W233$^{+\bullet}$ through charge recombination or further moves to a positive Arg residue (R286$^+$ or R338$^+$) to neutralize the Arg charge to destroy the salt bridge(s). To determine the dynamics of these charge recombination and charge neutralization, here we map out the W$^{+\bullet}$ temporal evolution (Fig. 2e) and thus also obtain the dynamics of W285$^{-\bullet}$. Figure 2e shows the total signal probed at 630 nm. The transient absorption decay of W285F (gray squres in Fig. 2e) agrees with its W* decay dynamics (the red curve) from the above fluorescence detection, excluding W$^{+\bullet}$ formation in the mutant. Conversely, the WT transient (gray circles in Fig. 2e) is clearly different from the W* dynamics in the pyramid center (the dark yellow curve in Fig. 2e), suggesting W$^{+\bullet}$ contribution in the signal. The total transient signal can be decomposed into two contributions of W* and W$^{+\bullet}$. The total W$^{+\bullet}$ signal appears in two rises and then one decay (Fig. 2e and Supplementary Fig. 7). The W$^{+\bullet}$ signal results from three possible reaction channels (Fig. 2f): W285$^{-\bullet}$/W233$^{+\bullet}$ charge recombination (CR), W285$^{-\bullet}$ electron transfer to neutralize positive Arg (ET), and W233$^{+\bullet}$ deprotonation (DP). By a model fitting of the three channels (see Supplementary Methods), we observed the charge recombination in 4 ps, the charge neutralization in 17 ps and the deprotonation in 3 ns (Fig. 2f). The total W233$^{+\bullet}$ signal can come from two components of W285$^{-\bullet}$/W233$^{+\bullet}$ and W285/W233$^{+\bullet}$ (inset of Fig. 2e and Supplementary Fig. 7). The former shows a rise and decay behavior, a reverse kinetics with slow formation (80 ps) and fast decay (~3.2 ps), leading to less accumulation of W285$^{-\bullet}$/W233$^{+\bullet}$ and thus a small amplitude of the initial 3.2-ps rise component. The latter shows an apparent rise mainly determined by 80 ps and then a decay (3 ns) of W285/W233$^{+\bullet}$. The amplitude of this latter component is proportional to the branching ratio of the ET channel to positive Arg, which is determined by the timescale of 17 ps. Thus, the productive efficiency ($E_{ET}$) of the Arg charge neutralization is 0.19. The QM/MM calculations showed a similar trend but with charge recombination in 0.4 ps and ET to arginine in 1.8 ps, faster by a factor of 10 (Supplementary Table 3). By multiplying the light-perception efficiency of 0.73 ($E_{QY}$)[20], charge-separation efficiency of 0.75 ($E_{CS}$) above and ET to Arg efficiency of 0.19 ($E_{ET}$) together, a total quantum yield of 0.104 was obtained, in excellent agreement with our steady-state measurement of dissociation quantum yield of 0.10 (Fig. 1d), suggesting that after W285$^{-\bullet}$-R286$^+$/R338$^+$ charge neutralization, all subsequent steps proceed to the final dimer dissociation.

**Charge separation direction and roles of nearby charged residues.** To further confirm our reaction mechanism of charge separation (W285$^{-\bullet}$-W233$^{+\bullet}$), charge neutralization (W285-R286$^{\bullet}$/R338$^{\bullet}$), and charge deprotonation (W233$^{\bullet}$) (Fig. 2f), we designed a series of mutants to solidify our observations (Supplementary Tables 1 and 2). For the charge-separation direction, we mutated the W285 and/or W233 to tyrosine (Y) to intentionally modulate ET direction between Y and W. All three

mutants show the faster fluorescence decay dynamics than the corresponding redox-inert phenylalanine (F) mutants (Supplementary Fig. 8a), implying charge-separation reactions occurring in the Y mutants, not in the F mutants, but with much slower rates than that in WT. Among the three mutants, W285Y and W285/233Y show unfavorable charge separation and a very minor response to extended UV-B + UV-C light irradiation, while W233Y exhibits the highest dissociation efficiency among the 3 mutants (Supplementary Fig. 8b), indicating that the charge does flow from the residue 233 to 285 as we observed in WT to disrupt the neighboring salt bridges. Y is a more unfavorable electron acceptor than W and thus having Y on 285 position hinders the formation of critical anionic radicals near R286/338. When we mutated R286$^+$ or R338$^+$ to neutral alanine (A) or glutamine (Q), both mutants even fail to form dimer after purification and the monomers show slower fluorescence decay dynamics than WT. But significantly, we mutated R286$^+$ or R338$^+$ with the same positively charged lysine (K) and the two mutants show the same behaviors as the wild type (Supplementary Fig. 8c, d), indicating the critical salt bridges for the dimer formation and the essential positive charges near W285. The mutations of other interfacial salt-bridge Arg residues (R146$^+$ and R200$^+$) do not have obvious effects on dimer formation or dissociation (Supplementary Table 2), indicating the key function of R286$^+$ and R338$^+$ as a decisive switch that turns on UVR8 monomerization. The mutations of charged residues of R234$^+$ to neutral (Q) or D129 to neutral (N) or positive (K) residues near W233 give the normal function, but less efficient monomerization with the slower fluorescence decay dynamics (Supplementary Fig. 8e, f), suggesting that the charge separation is modulated by the local electrostatic property and structural architecture; the destabilization of W233$^{+•}$ environment by changing from a negative residue to a neutral or positive residue slows down the charge-separation rate, leading to lower dissociation efficiency (Fig. 1d). Furthermore, the hydrogen bonding between D129$^-$ with W233 also anchors the key W233, leading to a rigid conformation favoring a directional electron transfer to W285 or a long fluorescence lifetime (12.6 ns) in the absence of W285. We also removed the whole hydrophobic pyramid cluster by the mutant of W285/233/94/337G and found that the mutant is a constitutive dimer (Supplementary Fig. 6) with no dissociation function upon UV irradiation, suggesting that the cation-π interactions proposed in the early report[11] is not essential for dimer formation and dissociation.

**Water channel opening, interface flooding, and dimer dissociation.** To follow subsequent dissociation steps after neutralization of the key salt bridge(s), we conducted long molecular dynamics (MD) simulations for neutralized R286$^•$, R338$^•$, and R234$^•$ and for a parent state (no modified residues) as a control. Although complete monomerization could take milliseconds, as recently reported by transient grating experiments[23], an apparent tendency to dissociation was observed in 2-μs trajectories of R286$^•$ and R338$^•$, but not for the parent state or neutralized R234$^•$ simulations. After R286$^+$ or R338$^+$ becomes neutral, water molecules flush into the dimer interface (Fig. 3a), weakening electrostatic interactions and causing a few Å increase in intersubunit distance (Fig. 3b). Underneath the pyramid cluster, a central water channel with a dimeter of 7 Å is formed by the seven blades of each β-propeller monomer (Supplementary Fig. 9). When monomerization is initiated by R286$^+$ or R338$^+$ neutralization, the hydrophobic gate formed by the pyramidal Trp residues on top of the water channel opens up, allowing channel water molecules to enter the interface and exchange with

interfacial waters (Fig. 3c). By examining MD snapshots, we found that the ground-state and neutralized R234$^•$ dimers are structurally stable with the closed hydrophobic gate, while the neutralized R286$^•$ and R338$^•$ dimers make large conformational changes. For the R286$^+$ or R338$^+$ neutralized forms, the interfacial salt-bridges and hydrogen-bond networks are disrupted and water moves in. The salt bridge between R286 and D96 breaks apart within one nanosecond, reorienting D96 towards R234 (Fig. 3d, e and Supplementary Fig. 10). Various interfacial water molecules are dislocated as the hydrogen-bond network reorients (Fig. 3d, e), which is consistent with an earlier report of the ejection of a structural water interacting with W233, W285, D96, and R286.[24] Subsequently, several interfacial salt bridges are interrupted, forming water-solvated ions (Fig. 3d, e and Supplementary Fig. 10). As more water molecules enters the interface, interfacial charged residues are solvated by water molecules to evolve toward dimer dissociation. The role of water in stabilizing separated charges revealed here also explains the recent observation by native mass spectrometry, showing that the UVR8 dimer does not photo-dissociate following UV-illumination in the gas phase.[29] The most prominent structural changes occur in blades 5 and 6, where W285 and W233 are located (Supplementary Fig. 11). Importantly, we observed partially unwinding of the β-propeller architecture in both neutralized R286$^•$ and R338$^•$ trajectories (Fig. 3f and Supplementary Fig. 11), consistent to the recent X-ray studies by freezing the intermediate structures[24]. Furthermore, for the neutralized R338$^•$ dimer, a mutual unwinding and partial dissociation of two monomers were observed (Fig. 3f). The Trp/Arg diradical after dissociation will be reconverted to the closed-shell ground state probably in a longer time and could also be addressed by EPR techniques. Based on our MD simulations, this diradical species can exist for hundreds of nanoseconds (Fig. 3a, b) to effectively trigger the monomerization.

## Discussion

Based on the above findings, our previous light-harvesting results[20], as well as the crystal structure of *Arabidopsis* UVR8, we paint out a complete process of UVR8 dimer dissociation upon UV perception (Fig. 4a–d). In this new mechanism, a delicate light-harvesting network funnels excitation energy to the pyramid center, doubling the light perception efficiency from 0.35 to 0.73 (Fig. 4a)[20]. Subsequently, 75% of the pyramid exciton population leads to directional charge separation to form W233$^{+•}$ and W285$^{-•}$ in 80 ps. Consequently, the critical transient W285$^{-•}$ has 19% possibility to neutralize the neighboring positive R286$^+$/R338$^+$ residue(s) in 17 ps while the remaining dominant reaction returns futile ground state through charge recombination in 4 ps. Overall, these multiple elementary reactions result in the total dissociation quantum yield of 0.1. The relatively long distance between R286$^•$/R338$^•$ and W233$^•$, after deprotonation of W233$^{+•}$, effectively locks the electron on neutral radical R286$^•$/R338$^•$ by preventing back electron transfer from R286$^•$/R338$^•$ to W233$^•$, leading to unity yield in successive elementary steps for dissociation. In summary, the proposed mechanism involves two sequential electron transfer steps: W233* to W285 and then W285$^{-•}$ to R286$^+$/R338$^+$, followed by W233$^{+•}$ deprotonation to yield W233$^•$/Arg$^•$ radical pair. With the neutralized R286$^•$/R338$^•$, this critical switch triggers a domino effect to unzip the strong interfacial interactions of many salt bridges and hydrogen bonds orchestrating by flooding the interface with channel/interfacial water molecules and subsequent collective protein motions, leading to dimer dissociation to interact downstream protein partners and thus activate the UV-protective mechanisms in cell.

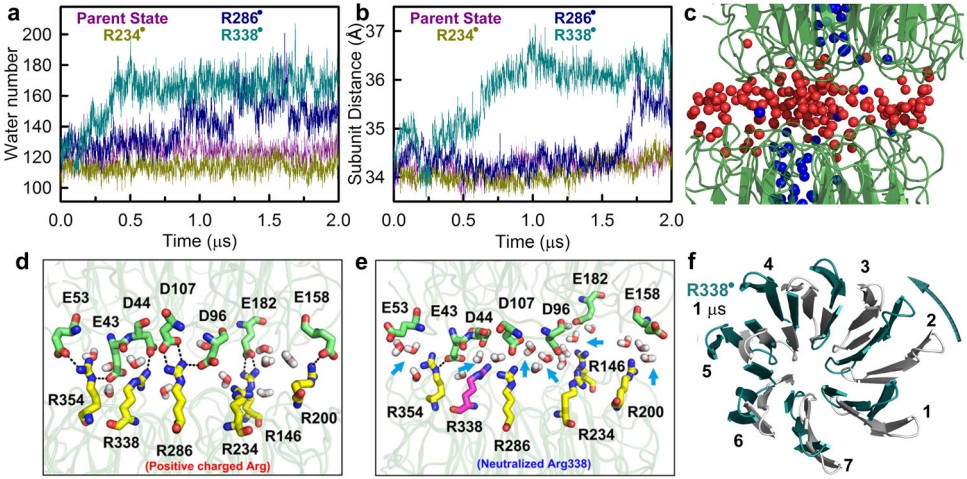

**Fig. 3 Computational simulations of dissociation. a** Number of water molecules at the dimer interface changes with time for the parent state, and neutralized R286•, R338•, and R234• from 2-μs simulations. **b** Inter-subunit distance (center-of-mass-distance) changes with time for the parent state, and neutralized R286•, R338•, and R234•. **c** A few channel water molecules (blue spheres) enter dimer interface via the opened gate after the hydrophobic cluster opens. Red spheres: interfacial water molecules that are not from the channel. **d** Interfacial salt-bridge interactions for the dark state of UVR8 dimer. **e** Salt-bridges and hydrogen-bond network was disrupted with water flooding in R338• neutralized state after 1 ns. Major changes in ion-pair interactions are indicated by light-blue arrows. **f** A typical snapshot from neutralized R338• simulation trajectory (cyan) after 1 μs is compared with a ground-state MD snapshot (gray). Two structures were aligned using monomer A and monomer B is shown here for comparison. The whole dimer structures are shown in Supplementary Fig. 11.

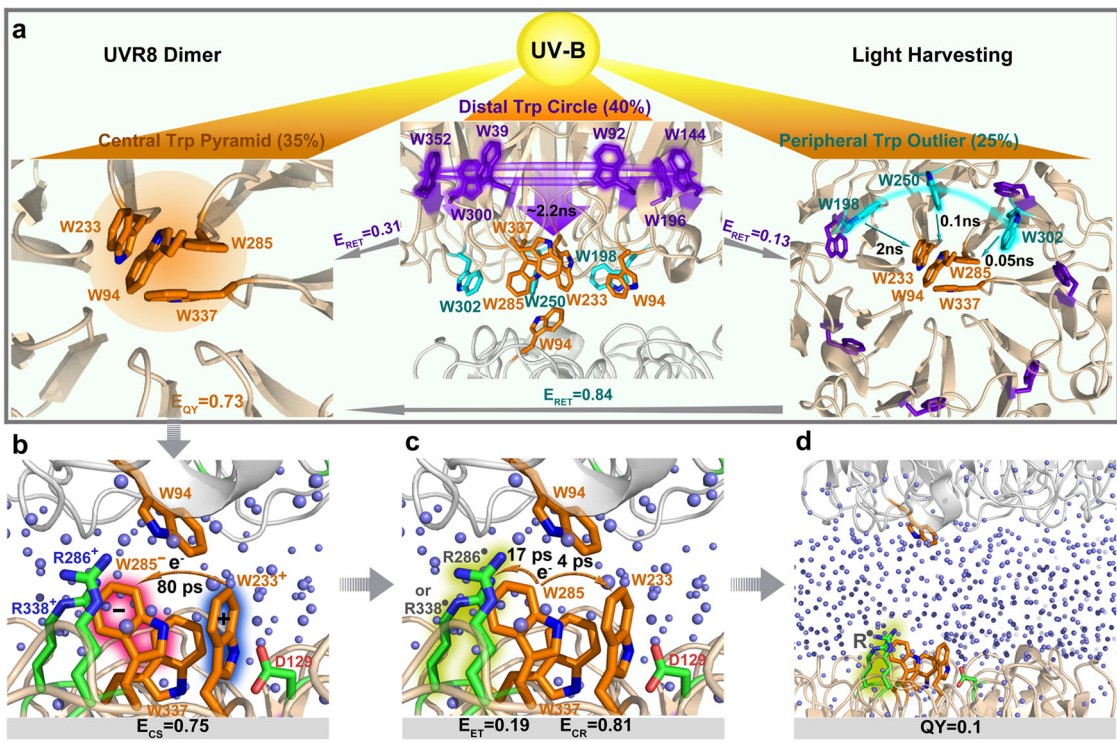

**Fig. 4 The molecular mechanism of UVR8 monomerization. a** Excitation energy is funneled to the pyramid center, enhancing light-perception efficiency from 0.35 to 0.73 ($E_{QY}$)[20]. Gray arrows denote directions of energy flows among 3 Trp groups, with labeled energy-transfer efficiency. The direct excitation percentages of the 3 Trp groups are shown in the brackets. **b** Electron transfer from W233 to W285 in 80 ps with an efficiency of 0.75 ($E_{CS}$). **c** The competing charge recombination (CR) in 4 ps and electron transfer (ET) from W285•- to R286+/R338+ in 17 ps with efficiency of 0.19 ($E_{ET}$). **d** Neutralized R286• or R338• leads to dimer dissociation in milliseconds[22] with a total quantum yield of 0.1.

## Methods

**UVR8 protein sample preparation and steady-state measurements.** The expression and purification of *Arabidopsis thaliana* UVR8 follows a reported procedure[11]. Briefly, the pET-29 (b) plasmid for UVR8 wild-type (WT) expression was generously provided by Prof. Yigong Shi from Tsinghua University. Mutant plasmids were constructed using site-directed mutagenesis kits (Qiagen) based on

the plasmid of UVR8 wild-type. The UVR8 ΔC plasmids were built by cloning truncated *A. thaliana* UVR8 (1–380 amino acids) genes into a pET-28 (a) vector using the NdeI and BamHI restriction enzyme sites. All DNA sequences are included in the Supplementary Methods. We sequenced the mutated DNA to ensure that the correct mutation(s) were introduced. Cell growth and protein purification of UVR8 ΔC follow the same protocol as the full-length UVR8.[11] The

plasmids were transformed to *E. coli* BL21-DE3 cells for protein expression. The cells were grown in Luria Broth (LB) at 37 °C to an OD of 1.0–1.2. After the cell culture was cooled down to 18 °C, 0.2 M IPTG was added to induce protein expression. The cells were pelleted following an induction period of 18–22 h at 18 °C. *E. coli* cells were resuspended in 150 mM NaCl, 25 mM Tris, pH 8.0, and lysed by sonication. After centrifugation for 1 h at ~23,000 × *g*, the supernatant was loaded into a Ni-Sepharose affinity column (Qiagen), and washed with 150 mM NaCl, 25 mM Tris, pH 8.0. The target protein was eluted by 250 mM imidazole, 25 mM Tris, pH 8.0. The product from the Ni column was further purified by a Source-15Q column (GE Healthcare) using gradient elution (from 0 to 1000 mM NaCl, 25 mM Tris, pH 8.0).

All samples were exchanged to a lysis buffer containing 150 mM NaCl, 25 mM Tris-HCl at pH 8.0 unless otherwise noted. For 290-nm excitation, we used protein samples with an optical density about 0.43 at 280 nm (about 4.7 μM dimer in 5-mm cuvette) to measure the steady-state absorption spectrum and an optical density about 0.086 at 280 nm (about 1 μM dimer in 5-mm cuvette) to obtain the steady-state fluorescence spectrum. For 310 nm and 315 nm excitations, samples with much higher protein concentrations (~50 μM dimer in 5-mm cuvette) were used for measurements. For experiments at different buffers ($D_2O$, phosphate), protein was exchanged to the desired buffer immediately before experiments with a pre-equilibrated desalting spin column (GE). The ingredients of $D_2O$ buffer: 150 mM NaCl, 25 mM Tris-HCl at pH 8.0 exchanged to deuterium oxide. Dimer/monomer state of proteins were determined by size exclusion chromatography (SEC) as reported previously[12]. UV-induced dissociation functions of protein samples were verified either by SEC[12] or by fluorescence intensity increase under continuous UV illumination[11,21].

**Dissociation quantum yield measurements**. The quantum yield of UVR8 dimer dissociation was measured as follows. We prepared a 125-μL lysis buffer (150 mM NaCl, 25 mM Tris-HCl at pH 8.0) containing 10 μM UVR8 dimer in a 5 mm × 5 mm square cuvette. The protein solution is then irradiated by a UV-B lamp (UVP, 6 W, ~1.4 mW/cm²) at a distance of 14 cm with at 300-nm longpass filter (UV-B, Supplementary Fig. 12 solid line). From the lamp spectrum and total power, we obtained photon flux at various wavelengths (Supplementary Fig. 12). The irradiated sample was then centrifuged at 21,036 × *g* for 5 min and loaded on a 10/30 superdex-200 column (GE Healthcare). The percentage of dimer dissociated was analyzed by calculating the ratio of areas of the dimer peak to the monomer peak with Gaussian function. For measuring the quantum yield of deuterated proteins, we stored the sample in the $D_2O$ buffer overnight (>10 h) before experiments.

**Fluorescence transient measurement with sub-nanosecond-resolved time correlated single photon counting (TCSPC)**. The sub-nanosecond-resolved time-correlated single photon counting (TCSPC) experiments were carried out in a Fluotime-200 system (PicoQuant). We used a commercially available LED with the center wavelength at 290 nm (PLS 290, Picoquant, ~1 μW) as the excitation source. Protein samples (about 5 μM dimer) were kept in a 5-mm cuvette during measurements. The instrument response function (FWHM 700 ps) was determined by measuring the scattering signal of UVR8 lysis buffer (150 mM NaCl, 25 mM Tris and pH = 8.0). The time window is 120 ns. No detectable dimer dissociation was observed under this experiment condition, as confirmed using size-exclusion chromatography. All transients were fitted with a multiple exponential decay model using the FluoFit software (PicoQuant).

**Fluorescence transient measurement with picosecond-resolved time correlated single photon counting (TCSPC)**. The picosecond-resolved time-correlated single photon counting (TCSPC) experiments were conducted with a Fluotime-200 system (PicoQuant). A Ti-sapphire oscillator (Tsunami, Spectra-physics) was used to generate fundamental frequency, which was then subject to third harmonic generation with a tripler (TPL fs tripler, minioptic) to obtain the 315 nm pump pulses (80 MHz, 0.2–0.4 μW). The instrument response function (FWHM 40 ps) was determined by measuring the scattering signal of UVR8 lysis buffer (150 mM NaCl, 25 mM Tris and pH = 8.0). Protein samples (about 20 μM dimer) were kept in a 5-mm cuvette for measurements and no detectable dimer dissociation was observed after the experiments. All transients were fitted with a multiple exponential decay model using the FluoFit software (PicoQuant).

**Femtosecond-resolved broadband transient absorption spectroscopy**. For all measurements, 290 nm or 315 nm pump pulses (1 kHz) were generated by a commercial optical parametric amplifier (TOPAS, Spectra-Physics). The pump pulse energy was attenuated to 80–100 nJ/pulse before being focused into UVR8 samples (4 mg/mL), which is kept in a 5-mm quartz cell with constant magnetic stirring. We generated white-light continuum (WLC) probe pulses by focusing the 800-nm pulses on a sapphire crystal with 2-mm thickness and the resulting broadband pulses were split into a probe beam and a reference beam. The probe beam was focused and overlapped with the pump beam at the sample cell. After that, probe and reference beams were focused into two optical fibers guiding the light into a monochromator (IHR320, Horiba). The incoming pulses were dispersed by a 300 grooves/mm grating and then imaged onto a 1024 × 256 pixel CCD detector (Synapse, Horiba) for data acquisition and analysis. The experiments

were done at the magic angle (54.7°) with ~350 fs instrument response. To further minimize the local photobleaching and monomer accumulation, the pump beam needs to be blocked for 1 s every 50 laser shots (50 ms). Otherwise the data will be contaminated with UVR8 monomer signal. For WT, samples were replaced with fresh samples before any observable dimer dissociation occurs, as checked by the superdex-200 size-exclusion column. For 315-nm excitation spectra, Savitzky-Golay smoothing (100 points of window) was conducted, and then the smoothed curves were averaged with raw data to give the data shown in Supplementary Fig. 5 and Fig. 2d.

**Femtosecond-resolved single wavelength transient absorption spectroscopy**. The experimental layout has been detailed elsewhere[30,31]. Briefly, for all measurements, pump and probe beams (1 kHz) were generated by two optical parametric amplifiers (TOPAS, Spectra-Physics). UVR8 protein samples (4 mg/mL) in a phosphate buffer (pH 7.5, 50 mM phosphate) were kept in a 5-mm quartz cell with constant magnetic stirring. The pump pulse energy was attenuated to 80–100 nJ/pulse before being focused into the sample cell. The instrument response time is about 250 fs and all experiments were done at the magic angle (54.7°). The pump beam needs to be blocked for 1 s every 50 laser shots (50 ms) to allow adequate equilibrium. For WT, samples were frequently replaced with fresh samples to ensure negligible monomer formation during experiments.

**Kinetic model fitting for transient absorption data at 315-nm excitation and $E_{ET}$ calculation**. The kinetic model is detailed in the Supplementary Methods. By model fitting, $\tau_{ET}$, $\tau_{CR}$ and $\tau_{DP}$ were obtained as 17 ps, 4 ps and 3 ns. Thus, branching ratio of ET ($E_{ET}$) was calculated as:

$$E_{ET} = \frac{(\tau_{ET})^{-1} + (\tau_{DP})^{-1}}{(\tau_{ET})^{-1} + (\tau_{CR})^{-1} + (\tau_{DP})^{-1}} = \frac{(17\,\text{ps})^{-1} + (3000\,\text{ps})^{-1}}{(17\,\text{ps})^{-1} + (4\,\text{ps})^{-1} + (3000\,\text{ps})^{-1}} = 0.19$$

(1)

**Numerical simulation of absorption transients at 290 nm excitation**. More details are described in the Supplementary Methods. 290-nm light can pump all 3 Trp groups: distal tryptophan ($6W_d$), peripheral tryptophan ($3W_p$) and pyramid center tryptophan ($4W_c$). We first simulated W excited state ($W^*$) temporal evolution for 3 groups with the energy transfer model as described in another study[20]. With $[W_{c1}]_t$, population evolution of $W^{+\bullet}$ was simulated using equations listed in Supplementary Methods.

**MD simulations**. The simulations of parent state (no modified residues), electron excited and transfer state were set up using CHARMM c38a2.[32] CHARMM27 force field[33] with CMAP correction (inclusion of an energy correction map)[34] was used for parent state calculation. The system was first subjected to energy minimization, followed by gradually heating from 10 K to 300 K every 10 K using 100 ps NPT simulations at each temperature. For simulations of the electron excited and transfer states, the corresponding patch residues ($W^*$, $W^{+\bullet}$, $W^{-\bullet}$, $W^\bullet$, and $R^\bullet$) were used, which the charges in the force field were modified from time dependent density functional theory (TDDFT) and multistate density functional theory (MSDFT)[35]. For atomic charges of radical pairs, gas phase MSDFT calculations were performed on residue pairs by PBE0 functional with HF correction for the off-diagonal Hamiltonian matrix element. Atomic charges of $W^*$ was acquired by time dependent range-separated hybrid functional TD-CAM-B3LYP. 6–31 + G(d) basis set was used for all QM calculations. All simulations were first carried out by using CHARMM during initial 20 ns equilibration and 20 ns production for further combined quantum mechanical and molecular mechanical (QM/MM) studies. Then the parent state, R338 neutralized, R286 neutralized and R234 neutralized state were selected for long time simulations up to 2 μs using GROMACS-4.6.5 molecular dynamics code[36] to study the mechanisms of dimer dissociations. The non-pair list was updated every 10 steps. The grid neighbor searching method was applied in the simulation with a 10 Å cutoff distance for the short-range neighbor list. Electrostatic interactions were treated by using the Partical-Mesh Ewald (PME) summation method[37] with a 14 Å for long range and 10 Å for short-range electrostatic cutoff, respectively. The short-range cutoff for van der Waals interactions during the simulation was 12 Å. The isotropic pressure coupling was achieved by Parrinello-Rahman method[38] with a compressibility of $4.5 \times 10^{-5}$ bar⁻¹. All simulations were carried out at 300 K in NPT ensemble and 1 atm pressure with a time step of 2 fs

**Count water number at the interface**. Three carbon-α (CA) atoms of D129, W285, and A24 for each monomer were selected to make two planes as shown in Supplementary Fig. 13. The interfacial area is between the planes. The geometric center of the interface may approximately equal to the center of mass for those six CA atoms. Then we counted the number of oxygen atoms of water both in between the 2 planes and within 18 Å distance of the center of mass of the 6 CA.

**QM/MM calculations**. The details about the theory and methods are available in the Supplementary Methods. Briefly, we extracted 100 snapshots (one snapshot every 200 ps) from the 20 ns production simulations of the parent state, electronic excited states and charge transfer states to statistically investigate the electron

transfer (ET) rate based on the QM/MM calculations. The side chains of residues in QM region were terminated between $C_\beta$ and $C_\alpha$ atoms with a hydrogen link atom. The corresponding backbone and other residues were treated as MM region. The single excited state of residue was acquired by time dependent range-separated hybrid functional TD-CAM-B3LYP.[39] MSDFT calculations were performed by PBE0 functional[40] with HF correction for the off-diagonal Hamiltonian matrix element. $6-31 + G(d)$ basis set was used for all the calculations. All the QM/MM calculations were performed with a locally modified version of GAMESS code[41] in CHARMM quantum part. MSDFT was used to construct the diabatic states of electronic localized excitation and electron transfer. TD-CAM-B3LYP together with MSDFT was used to obtain the locally excited W233. In addition, MSDFT was used to calculate electronic coupling matrix element between two diabatic states for rate constants. The reorganization energies and driving forces are computed on the basis of linear response theory from molecular dynamics simulations of the initial and final diabatic states for each electron transfer reaction, and the electronic coupling matrix elements were averaged over 100 configurations during 20 ns molecular dynamics simulations (Supplementary Table 3 and Supplementary Fig. 14). The free energy barriers were determined according to Marcus-Hush theory for electron transfer.

**Reporting summary**. Further information on research design is available in the Nature Research Reporting Summary linked to this article.

## Data availability
The data that support the findings of this study are available from the corresponding author upon reasonable request. Source data are provided with this paper.

## Code availability
All relevant computer codes supporting this study are available from the corresponding author upon reasonable request.

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

## Acknowledgements
We thank Prof. Yigong Shi (Tsinghua University) for generously providing the UVR8 plasmid. This work was supported in part by the National Institute of Health (Grant GM144047 to D.Z. for experiments and GM46736 to J.G. for computation) and the National Natural Science Foundation of China (for support of collaboration efforts through a visit of XL and summer stays of D.Z. in Shanghai Jiao Tong University and Grant No. 21533003 to J.G. for support of H.R. to complete computational work).

## Author contributions

D.Z. designed the research. X.L., M.K., Z.L., F.Z., and L.W. performed the experiments. H.R. and J.G. did computational studies. D.Z. and X.L. wrote the paper. All authors discussed and edited the manuscript.

## Competing interests

The authors declare no competing interests.

## Additional information

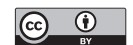

