## [Peer Review File · Nature Communications]

Dynamics and mechanism of dimer dissociation of photoreceptor UVR8REVIEWER COMMENTS

Reviewer #1 (Remarks to the Author):

Comments to authors

This contribution describes a very detailed experimental / computational study of the photodissociation mechanism of the plant photoreceptor UVR8. Although several hypotheses have been tested in the literature, both experimentally and computationally, few have proved conclusive. Overall, I find the data and analysis presented here the most compelling to date. That said, the authors fail to discuss their data in the context of the literature in anywhere near enough detail, which makes it difficult for the reader to judge the study's merits against previous efforts. In doing so they also fail to give due credit to past findings that agree with, and preceded, their own. To address this, they must conduct a substantial rewrite before I can recommend publication.

Major points

Important matters that have been largely over-looked include, but are not limited to:

1. *Page 2, lines 61-62* – "...and several hypotheses are highly controversial." Although the authors are clearly aiming for brevity in the introduction, they should summarise these hypotheses here and why they are controversial. This would better set the scene.
2. A study using quantum chemical calculations [Proc. Natl. Acad. Sci. U. S. A. 111, 5219-5224 (2014)] already concluded that charge separation between W285 and W233 is likely. The observation of charge separation in the current study is not novel, therefore, and instead provides experimental evidence in support of this mechanism. The authors should therefore discuss their data (in detail) in the context of this study.
3. As the authors point out, a proton-coupled electron transfer mechanism was previously proposed for UVR8 [*J. Am. Chem. Soc.* **137**, 8113-8120 (2015)]. They only mention this is passing, however, and do so to dismiss it. I don't necessarily disagree with their conclusions but think a more detailed discussion is necessary for the case to be convincingly made, especially for the general reader.
4. The authors present MD simulations that convincingly support the involvement of water molecules in dimer dissociation, suggesting they enter the dimer interface and disrupt the cross dimer salt bridges. Again, this isn't a novel observation:
 - Native mass spectrometry showed that the UVR8 dimer does not photo-dissociate following UV-illumination in the gas phase, and it was concluded that this was almost certainly because water was needed to stop the salt-bridges reforming [Proc. Natl. Acad. Sci. USA, 116(4), 1116-1125 (2019)]. This paper isn't cited at all and should be discussed in detail in light of the authors' findings that support the original proposal.
 - Although the authors do cite a relevant dynamic crystallography study [Nat. Plants 1, 14006 (2015)], they do so only to state that their findings are consistent with observations that the UVR8 monomers partially unwind. They miss out the other main conclusion from that study, that a stabilising water is ejected from the dimer interface as part of the photo-dissociation mechanism. Surely if data are presented to support a role of water, they should be discussed with detailed reference to these findings also.

On a separate note:

5. *Page 3, lines 83-84* – “In deuterium oxide (D₂O), the QY value is similar, suggesting that the proton transfer does not involve in critical elementary reactions during dimer dissociation.” This statement assumes that the residues that might be involved in any putative proton transfer are solvent accessible to enable H/D exchange. Do the authors have evidence for this? Indeed, do their conclusions that some of the Trps from the pyramid form a hydrophobic gate to this area, stopping waters from entering, suggest not? These apparently conflicting observations need to be reconciled for either conclusion to stand up to scrutiny.

Minor points

6. The English needs tightening up throughout
7. *Abstract*
 - “Photoreceptors usually exist in form of dimer in nature and their monomerization initiates biological functions.” is a strange opening line that suggests this to true of all photosensory protein, which it is not. It should be rewritten.
 - The following line is too strongly stated and should be tempered along the following lines “However, the dissociation mechanism is still ~~unknown~~ poorly understood.”
 - The sentence starting “Here, ...” is far too long and should be broken up.
8. *Page 2, lines 61-62* – Again, the following line is too strongly stated and should be tempered
“However, the UV-induced dissociation mechanism is still ~~unknown~~ poorly understood ...”
9. *Page 5, lines 126-128* – Although I’m pretty sure I know what they are trying to say, the following line is expressed oddly and should be edited along the following lines:
“W285, flanked by two neighboring positively charged arginine residues of R286+ and R338+, is more likely to be an electron acceptor ~~from to form~~ W285- while W233 is the electron donor ~~and formsef~~ W233+,...”
10. *Page 7* – for the general reader, perhaps the authors should more clearly state the rational for the W to Y mutations in the context of their relative redox potentials and the impact this is likely to have on the electron transfer dynamics.
11. *Fig 3c* – the authors need to make it clear in the figure caption what the red spheres are.
12. *Supplementary Fig. 9d* – the way the data are plot in this figure make it very difficult to discern whether or not there are differences between the different variants. It should be plot more clearly.

Reviewer #2 (Remarks to the Author):

The manuscript describes the light-induced dissociation of the UVR8 protein-dimer into monomers. The tools used are spectroscopy, site-directed mutagenesis and computer simulations. Eventually, a mechanism which involves two Trp in the so-called pyramid is derived. It starts with a charge separation, leads to an electron transfer and then a series of larger changes which leads to a water influx causing the dissociation.

The mechanism is plausible. In particular the simulation methods are state of the art. The sampling of 100 snapshots will provide reasonable statistics. And the fact that the critical steps are in the range from ps to ns leaves no choice but to use the rate equation, instead of explicit dynamics. However, some clarification of the methodology is required:

- 1) The QM region in the QM/MM simulation is not clearly defined. Where is the boundary between the QM and MM region placed? What is the type of embedding?
- 2) Why are aspartates included in the QM region? It is not explained in the SI.
- 3) What is the purpose of using MSDFT? This is not clear. Are the results of MSDFT compared to TD-CAM-B3LYP?
- 4) How are the locally excited tryptophanes obtained? This is critical for the proposed mechanism. TD-DFT tends to delocalize the excitation. So are the authors taking one tryptophane by one and then construct the wavefunction from block-localized orbitals?

In addition, I recommend to replace "ground state" in the section "Water channel opening, interface flooding and dimer dissociation" by "dark state" or "parent state".

Reviewer #3 (Remarks to the Author):

This is an important paper that opens a new research line in mechanistic photobiology. A substantially complete mechanism going from photon absorption up to a signaling protein dimer dissociation is proposed on the basis of experimental (mutational and spectroscopic) and computational (structural/dynamic and photochemical) studies. Both the corresponding authors are well known experts in their respective fields.

The mechanism for the activation of the UVR8 photoreceptor of plants is found to involve the following steps: light absorption (in part studied earlier) leading to exciton formation at a kind of "primitive" reaction center formed by a loose tryptophan dimer, electron transfer (charge separation) within the dimer with formation of a radical-cation - radical anion pair, a second electron-transfer reaction of the radical anion with an arginine leading to an arginine radical (neutral), disruption of salt-bridges, proton transfer recovering the neutrality of the reaction center, water influx, monomer separation. This is a novel, temporally and spatial multi-scale and, as far as I can tell, unique mechanism certainly suitable for publication in Nature Communication.

The manuscript must be improved before final acceptance. Here is a list of recommendations:

General comment: at the end of the reactive processes the system generates a diradical R286[•]/R338[•] and W233[•] but it is not clear how this diradical is then necessarily reconverted to the stable neutral closed-shell species. The authors should add some explanation or hypothesis here.

Specific comments:

1) Page 2, line 62. The reader would be interested to know what are the main hypothesis previously proposed for the activation process. In the conclusions the authors should also summarize the most credited one and explain why, according to the present study, those hypotheses are not valid.

2) Page 2, line 62. "a core ring". The precise composition of the three groups of Trp residues has to be better defined or rephrased. For instance it is not clear what is the first group forming a core ring and where exactly they are located (Fig. 1b is unfortunately not very helpful). Is this "core" comprising the three groups? The description is presently quite confusing. Also, the pyramid has to be better defined. From the description it seems that a pyramid with triangular faces would be better defined as and elongated tetrahedron. Is it a pyramid or a tetrahedron?

3) Page 3, line 94. Is the "cluster" and the "pyramid" (or tetrahedron) the same system? One should use the same definition for the same molecular framework. I believe that a scheme is necessary here illustrating schematically the structural features described in words. The reader may get confused without a scheme and then have issues in following the mechanism clearly.

4) Page 4, line 98. Now the authors use the words "pyramid-cluster". They should stick to the same wording.

5) Page 4, line 102. It is not clear why W33 would be an energy sink. It is apparently an electron donor. I would say that the energy sink (i.e. in terms of reaction energy) is the W285-W233 dimer.

6) Page 4. Several places (this comment is extended to the full paper). It is important to use the correct symbols and naming for reaction intermediates. So, after the ET reaction W285 is a radical-anion and W233 is a radical-cation. So W^+ is not correct but it has to be $W^{+\cdot}$ and not W^+ . There are other cases of inconsistencies throughout the manuscript.

7) Page 5. The mechanism requires two sequential electron transfers $W285 \leftarrow W233$ and then $R \leftarrow W285$. This has to be better clarified. In a sense a general scheme (may be in the Conclusion section) with the different parts of the proposed mechanism would greatly help the reader.

8) Page 6, line 148-150. This sentence has to be rephrased or corrected to improve readability

9) Page 6, line 151. It seems that "one" should be two and "two" should be one.

10) Page 9, line 212. It is not clear how the two R286 and R338 radical molecular mechanics parameters have been generated.

11) Page 9, line 222. The large conformation changes need to be briefly described in the main text.

12) Page 10. Conclusion section. As already mentioned above a general scheme of the entire mechanism should be given.

13) Page 15, line 355. CMAP has to be explained.

14) Page 15, line 358-359. What "patch residues" mean? Is the only parametrization done that of the point charges? Which particular charges have been generated using the QM calculation? Please, explain (in more details in the Supporting information).

15) Page 15, line 360. Have the use of TDDFT and MSDFT (with the functionals used in the paper) been benchmarked for excitation energy and charge transfer descriptions? Explain carefully for the general readership. Have TDDFT and MSDFT been used for excitation energies and for the description of the

charge transfer state respectively? Explain better (very little details are given in the supporting information on these core issues).

16) Pag. 4 Supporting Information. The diabaticization process is not clear enough. a and b are residues or monomers? Please, explain exactly to which particular moieties the wave functions refer to.

Reply to reviewers

Reply to Reviewer #1:

Comments to authors

This contribution describes a very detailed experimental/computational study of the photodissociation mechanism of the plant photoreceptor UVR8. Although several hypotheses have been tested in the literature, both experimentally and computationally, few have proved conclusive. Overall, I find the data and analysis presented here the most compelling to date. That said, the authors fail to discuss their data in the context of the literature in anywhere near enough detail, which makes it difficult for the reader to judge the study's merits against previous efforts. In doing so they also fail to give due credit to past findings that agree with, and preceded, their own. To address this, they must conduct a substantial rewrite before I can recommend publication.

Response: We thank reviewer 1 for his/her positive comments. We agree that this is the most comprehensive study on UVR8 dissociation mechanism to date. We have included significantly more discussions in the context of the literature to address the reviewer's concerns.

Major points

Important matters that have been largely over-looked include, but are not limited to:

1. Page 2, lines 61-62 – "...and several hypotheses are highly controversial." Although the authors are clearly aiming for brevity in the introduction, they should summarize these hypotheses here and why they are controversial. This would better set the scene.

Response: This is a great suggestion. There are at least 5 hypotheses in literature. Some are backed up partially by computational or experimental results, while others are proposed based on indirect evidence like the X-ray structure. The debate is mainly on the initial photochemical mechanism that triggers UVR8 monomerization. At line 61-67, we summarized most influential hypotheses to date: "Although all previous models suggest proton-coupled electron transfer (PCET) from W285 to R286."

2. A study using quantum chemical calculations [Proc. Natl. Acad. Sci. U. S. A. 111, 5219-5224 (2014)] already concluded that charge separation between W285 and W233 is likely. The observation of charge separation in the current study is not novel, therefore, and instead provides experimental evidence in support of this mechanism. The authors should therefore discuss their data (in detail) in the context of this study.

Response: We thank the reviewer for this suggestion. There are two major differences between our mechanism and the one proposed in the PNAS paper. 1) The PNAS paper proposed

W285⁺W233⁻ charge separation whereas we propose W285⁻W233⁺, a completely opposite charge-transfer direction; 2) The PNAS paper suggested the electrostatic perturbations by W285⁺W233⁻ drives proton transfer from Arg to Asp, neutralizing charges on the salt-bridges. But from our results, electron transfer from W285⁻ to Arg after charge separation is the key step to break inter-subunit salt-bridges and electrostatic interaction alone is not enough to drive monomerization (Supplementary Fig. 6). We have added detailed comparison between their theoretical model and our model in our paper at line 142-148: “The charge-transfer (CT) direction here is opposite to a previous computational study between W285 and W233 is essential to driving dimer dissociation.”

3. As the authors point out, a proton-coupled electron transfer mechanism was previously proposed for UVR8 [J. Am. Chem. Soc. 137, 8113-8120 (2015)]. They only mention this is passing, however, and do so to dismiss it. I don't necessarily disagree with their conclusions but think a more detailed discussion is necessary for the case to be convincingly made, especially for the general reader.

Response: This is a very good suggestion. This JACS paper proposed W285 triplet state is formed via intersystem crossing. PCET from triplet W285 to nearby R286/D96 subsequently produces R286 neutral radical and protonated neutral D96, leading to disruption of this critical salt-bridge. However, with selective excitation at 315 nm, our data clearly indicates W233/W285 charge separation in 80 ps, accompanied by W⁺ formation. Since the authors of the JACS paper used non-selective 266-nm excitation, the triplet state observed in the JACS paper is mainly from distal and peripheral Trp, not from the Trp pyramid. As shown in supporting information of the JACS paper (Fig. S9 of JACS paper), WT and W285/233/337F have similar Trp triplet absorption peaked at 480 nm. We have added the following discussion from line 129: “Recent experiments suggest formation of W285 triplet state since the triplet absorption was also observed in the mutant of W233/285/337F.”

4. The authors present MD simulations that convincingly support the involvement of water molecules in dimer dissociation, suggesting they enter the dimer interface and disrupt the cross-dimer salt bridges. Again, this isn't a novel observation:

o Native mass spectrometry showed that the UVR8 dimer does not photo-dissociate following UV-illumination in the gas phase, and it was concluded that this was almost certainly because water was needed to stop the salt-bridges reforming [Proc. Natl. Acad. Sci. USA, 116(4), 1116-1125 (2019)]. This paper isn't cited at all and should be discussed in detail in light of the authors' findings that support the original proposal.

Response: The reviewer raised a good point. This PNAS paper provides important conformational insights into UVR8 monomerization using mass spectroscopy. But the water role that the PNAS raised is different from what we proposed in the manuscript. Here, we found water molecules play critical roles in weakening the cross-dimer electrostatic interactions during

dissociation. Water could solvate charged residues after the inter-subunit salt-bridges are interrupted, stabilizing the monomeric state. We do not agree that water was needed to stop the salt-bridge reforming because in solution, two monomers after dissociation will form dimer again. Nevertheless, we have cited the PNAS paper (reference 29) and added the following sentences at line 239-244: “As more water molecules enters the interface, does not photo-dissociate following UV-illumination in the gas phase.”

o Although the authors do cite a relevant dynamic crystallography study [Nat. Plants 1, 14006 (2015)], they do so only to state that their findings are consistent with observations that the UVR8 monomers partially unwind. They miss out the other main conclusion from that study, that a stabilizing water is ejected from the dimer interface as part of the photo-dissociation mechanism. Surely if data are presented to support a role of water, they should be discussed with detailed reference to these findings also.

Response: This is another important paper on UVR8 UV-induced monomerization. By freezing UVR8 at different temperatures under UV irradiation, intermediate structures during the monomerization process were obtained. They found that a structural water, interacting with W285, W233, R286 and D96, is ejected after photo excitation. In our MD simulations, water networks at the dimer interface are greatly perturbed after cross-dimer salt-bridges are broken. As a result, various water molecules near R286/R338, including the proposed one, are dislocated (Fig. 3d and 3e), which is consistent with the Nat. Plant paper. However, we don't think ejection of this particular water molecule is a critical step of dimer dissociation. It's one of many consequences caused by R286/R338 charge neutralization and part of the disruption of the overall hydrogen-bond network. We have included the following discussion at line 236 “Various interfacial water molecules are dislocated as the hydrogen-bond network reorients (Fig. 3d, e), which is consistent with earlier report of the ejection of a structural water interacting with W233, W285, D96 and R286.”

On a separate note:

5. Page 3, lines 83-84 – “In deuterium oxide (D₂O), the QY value is similar, suggesting that the proton transfer does not involve in critical elementary reactions during dimer dissociation.” This statement assumes that the residues that might be involved in any putative proton transfer are solvent accessible to enable H/D exchange. Do the authors have evidence for this? Indeed, do their conclusions that some of the Trps from the pyramid form a hydrophobic gate to this area, stopping waters from entering, suggest not? These apparently conflicting observations need to be reconciled for either conclusion to stand up to scrutiny.

Response: For the H/D experiment, we stored the protein in the D₂O buffer overnight (>10h) before measuring the quantum yield. We were assuming tryptophan N_{e1} and various interfacial charged residues (Arg, Asp, Glu and Lys) are deuterated. As shown in Fig. 1b, those tryptophan residues are accessible by interfacial waters. The labeling efficiency could be examined in the

future by mass spectroscopy. To clarify this, we have added the following sentence in the methods: “For measuring the quantum yield of deuterated proteins, we stored the sample in the D₂O buffer overnight (>10h) before experiments.”

The main purpose of this experiment to show that proton transfer is not involved in critical branching steps. Although H/D exchange could slow down W233⁺ deprotonation, it won't affect dissociation quantum yield based on our model.

MD simulations show that initially the tryptophan pyramid blocks water entering from the hole and after charge separation, the hole is open and water from the hole can enter interface, besides the neighboring water molecules flood into the interface.

Minor points

6. The English needs tightening up throughout

Response: We have modified several sentences according to the reviewer's suggestions (see below and the revised manuscript).

7. Abstract

o “Photoreceptors usually exist in form of dimer in nature and their monomerization initiates biological functions.” is a strange opening line that suggests this to true of all photosensory protein, which it is not. It should be rewritten.

Response: We have rewritten this sentence to “Photoreceptors are a class of photo-sensing proteins with critical biological functions.”

o The following line is too strongly stated and should be tempered along the following lines “However, the dissociation mechanism is still unknown poorly understood.”

Response: We have changed the word “unknown” to “poorly understood”.

o The sentence starting “Here, ...” is far too long and should be broken up.

Response: This is a good suggestion. We have broken this sentence into two sentences.

8. Page 2, lines 61-62 – Again, the following line is too strongly stated and should be tempered “However, the UV-induced dissociation mechanism is still poorly understood ...”

Response: We agree with the reviewer. We have changed the word “unknown” to the phrase “poorly understood”.

9. Page 5, lines 126-128 – Although I'm pretty sure I know what they are trying to say, the following line is expressed oddly and should be edited along the following lines: "W285, flanked by two neighboring positively charged arginine residues of R286⁺ and R338⁺, is more likely to be an electron acceptor from to form W285⁻ while W233 is the electron donor and forms of W233⁺,..."

Response: Thank the reviewer for this suggestion. We have replaced the original sentence with the following one "W285 is flanked by two neighboring positively charged arginine residues of R286⁺ and R338⁺, and thus is more likely to be an electron acceptor; W233 could be the electron donor since W233⁺ is stabilized through a hydrogen bond with the negatively charged D129."

10. Page 7 – for the general reader, perhaps the authors should more clearly state the rationale for the W to Y mutations in the context of their relative redox potentials and the impact this is likely to have on the electron transfer dynamics.

Response: A key intermediate in the proposed mechanism is W285^{•-} anionic radical, which neutralizes Arg286/338. The gas phase electron affinity (EA) of indole ring is 3.5 meV (Carles, S. *et. al.*, J. Chem. Phys., 2000, 112, 8, 3726-3734), suggesting W^{•-} is possible. To the best of our knowledge, there is no report of electron affinity of phenol ring, indicating that Y^{•-} is very unstable. For a W/Y pair, electron-transfer direction should be from Y to W, producing Y⁺/ W^{•-} biradical. Thus, the W285Y mutant shows negligible dissociation because W233⁺/Y285^{•-} cannot be formed or is unstable. W233Y still show weak dissociation since Y233⁺/W285^{•-} could be formed. We have modified a sentence at line 196 to "Y is a more unfavorable electron acceptor than W and thus having Y on 285 position hinders the formation of critical anionic radicals near R286/338."

11. Fig 3c – the authors need to make it clear in the figure caption what the red spheres are.

Response: In Fig. 3c was obtained by comparing two adjacent MD snapshots (snapshot1, snapshot2, 100 ps away) when the channel is open. The figure shows structure of snapshot2. If an interfacial water molecule in snapshot2 was found in the channel in the earlier snapshot1, the molecule was considered as moving from the channel and colored blue. Other interfacial molecules were shown in red. Thus, blue spheres are water molecules from the channel; red spheres are other molecules that are not from the channel. We have included the following sentence in figure caption: "Red spheres: interfacial water molecules that are not from the channel."

12. Supplementary Fig. 9d – the way the data are plot in this figure make it very difficult to discern whether or not there are differences between the different variants. It should be plot more clearly.

Response: As shown in Supplementary Fig. 9d, even in R286* (blue line) or R338* (red line) trajectories, the gate is closed most of the time (>95%). The spikes in the blue and red curves are the open state. That is the reason the four trajectories are highly similar. It seems most water molecules enter the dimer interface from protein outside rather than the channel.

Reply to Reviewer #2:

The manuscript describes the light-induced dissociation of the UVR8 protein-dimer into monomers. The tools used are spectroscopy, site-directed mutagenesis and computer simulations. Eventually, a mechanism which involves two Trp in the so-called pyramid is derived. It starts with a charge separation, leads to an electron transfer and then a series of larger changes which leads to a water influx causing the dissociation.

The mechanism is plausible. In particular the simulation methods are state of the art. The sampling of 100 snapshots will provide reasonable statistics. And the fact that the critical steps are in the range from ps to ns leaves no choice but to use the rate equation, instead of explicit dynamics. However, some clarification of the methodology is required:

Response: We kindly appreciate the referee's favorable comments on the rationality of the computational method using in this work. The following is a summary of the point-by-point responses and changes that have been made according to reviewer's comments.

1) The QM region in the QM/MM simulation is not clearly defined. Where is the boundary between the QM and MM region placed? What is the type of embedding?

Response: The QM/MM simulation is used to compute rate constants for the initial electron transfer reactions from the photochemically excited W233* to W285, W233⁺ W285⁻ charge recombination, and the subsequent charge propagation to neutralize either R286 or R338. The QM part includes side chains of W285/W233, W285/W233/D129, W285/R286/D97/D107 and W285/R338/E43/D44 for calculating W285/W233 charge separation, charge recombination, ET to R286 and ET to R338, respectively, as described in QM/MM calculations of SI. The side chains of residues in QM region were terminated between C_β and C_α atoms with a hydrogen link atom. The corresponding backbone and other residues are treated as MM region. The type of embedding is the link atom approach [1], which has been widespread use in QM/MM calculations.

To make clear, the following sentences were integrated into the QM/MM calculations part of the revised Methods section and SI.

“The side chains of residues in QM region were terminated between C_β and C_α atoms of with a hydrogen link atom. The corresponding backbone and other residues were treated as MM region.”

[1] Singh UC, Kollman PA (1986) J Comput Chem 7:718–730

2) Why are aspartates included in the QM region? It is not explained in the SI.

Response: The charged residue of E43, D44, D97, D107, and D129 were treated as QM region due to they are very close to the key residue of W233, W285, R286 and R338 in active sites,

which has a significant quantum effect. The classically molecular mechanics (MM) cannot give good description of their interaction.

To make clear, the following sentences were integrated into the QM/MM calculations part of the revised SI.

“The charged residue of E43, D44, D97, D107, and D129 were treated as QM region because they are very close to the key residue of W233, W285, R286 and R338 in active sites, which has a significant quantum effect.”

3) What is the purpose of using MSDFT? This is not clear. Are the results of MSDFT compared to TD-CAM-B3LYP?

Response: MSDFT was used to construct the diabatic states of electronic localized excitation and electron transfer. TD-CAM-B3LYP together with MSDFT was used to obtain the locally excited W233. In addition, MSDFT was used to calculate electronic coupling matrix element for rate constants.

To make clear, the following sentences were integrated into the QM/MM calculations part of the revised Methods section and SI.

“MSDFT was used to construct the diabatic states of electronic localized excitation and electron transfer. TD-CAM-B3LYP together with MSDFT was used to obtain the locally excited W233. In addition, MSDFT was used to calculate electronic coupling matrix element between two diabatic states for rate constants.”

4) How are the locally excited tryptophans obtained? This is critical for the proposed mechanism. TD-DFT tends to delocalize the excitation. So are the authors taking one tryptophane by one and then construct the wavefunction from block-localized orbitals?

Response: The reviewer is absolutely correct for his/her comments. The excited states directly obtained by TDDFT are based on the Born-Oppenheimer approximation, which gives rigorously adiabatic excited states. To construct locally excited W233, we employ MSDFT, in which the exciton wave functions of W233 are localized within the molecular fragment, to quantify the energies of excited state as shown in equation 16 of SI.

5) In addition, I recommend to replace "ground state" in the section "Water channel opening, interface flooding and dimer dissociation" by "dark state" or "parent state".

Response: Thanks for the referee's constructive suggestions. The “parent state” was used to replace various “ground state” in the revised manuscript and SI. We also changed the figure legend in Fig. 3a and b.

Reply to Reviewer #3:

This is an important paper that opens a new research line in mechanistic photobiology. A substantially complete mechanism going from photon absorption up to a signaling protein dimer dissociation is proposed on the basis of experimental (mutational and spectroscopic) and computational (structural/dynamic and photochemical) studies. Both the corresponding authors are well known experts in their respective fields.

The mechanism for the activation of the UVR8 photoreceptor of plants is found to involve the following steps: light absorption (in part studied earlier) leading to exciton formation at a kind of "primitive" reaction center formed by a loose tryptophan dimer, electron transfer (charge separation) within the dimer with formation of a radical-cation - radical anion pair, a second electron-transfer reaction of the radical anion with an arginine leading to an arginine radical (neutral), disruption of salt-bridges, proton transfer recovering the neutrality of the reaction center, water influx, monomer separation. This is a novel, temporally and spatial multi-scale and, as far as I can tell, unique mechanism certainly suitable for publication in Nature Communication.

Response: We appreciate the reviewer 2 for his/her very positive comment. As exactly the reviewer said, the proposed mechanism in this paper is novel and deserves to be published in Nature Communications.

The manuscript must be improved before final acceptance. Here is a list of recommendations:

General comment: at the end of the reactive processes the system generates a diradical R286·/R338· and W233· but it is not clear how this diradical is then necessarily reconverted to the stable neutral closed-shell species. The authors should add some explanation or hypothesis here.

Response: The reviewer raised a good point. The exact reversing mechanism is not clear, which could be addressed by future studies, especially with EPR technique. However, in biological systems, these diradicals can get recovered by taking electron or proton from solution. Although the whole monomerization process takes milliseconds, the lifetime of the diradical could be shorter, *e.g.*, on microsecond time scales. At some point, monomerization becomes irreversible even after the diradical is reconverted to the closed-shell ground state. From our MD trajectory, it takes at least hundreds of nanoseconds for the inter-subunit distance to increase, providing a lower limit of the lifetime. We added the following content at line 250: "How the Trp/Arg diradical is reconverted to to effectively trigger the monomerization."

Specific comments:

1) Page 2, line 62. The reader would be interested to know what are the main hypothesis previously proposed for the activation process. In the conclusions the authors should also summarize the most credited one and explain why, according to the present study, those hypotheses are not valid.

Response: The reviewer provides an excellent suggestion. There are at least 5 hypotheses, either backed up by partial computational/experimental results or based on the X-ray structure. The debate is mainly on the initial photochemical mechanism that triggers UVR8 monomerization. At line 61, we summarized most influential hypotheses to date: “Although all previous models suggest proton-coupled electron transfer (PCET) from W285 to R286.” We also added detailed discussions in line 128-134 and 142-148, stating the reason why other hypotheses are not valid.

2) Page 2, line 62. “a core ring”. The precise composition of the three groups of Trp residues has to be better defined or rephrased. For instance it is not clear what is the first group forming a core ring and where exactly they are located (Fig. 1b is unfortunately not very helpful). Is this “core” comprising the three groups? The description is presently quiet confusing. Also, the pyramid has to be better defined. From the description it seems that a pyramid with triangular faces would be better defined as and elongated tetrahedron. Is it a pyramid or a tetrahedron?

Response: The reviewer’s suggestion is very good. “A core ring” refers to the 6 distal tryptophan in the middle of the protein, which forms a symmetrical ring from top view (see reference 20). The word “pyramid” was first used in the 2012 structural paper (Christie, J. M. *et al.* Science 335, 1492-1496, 2012). The authors used “tryptophan pyramid” or “Trp pyramid” extensively in the paper to describe the clustered Trp tetramer by W233, W285, W337 and W94b. The field has accepted this definition and used it in literature. Thus, it is better for us to stick to the convention and name this tetramer a “pyramid”. We have included the residues numbers when introducing the three Trp groups to help general readers to understand.

3) Page 3, line 94. Is the "cluster" and the "pyramid" (or tetrahedron) the same system? One should use the same definition for the same molecular framework. I believe that a scheme is necessary here illustrating schematically the structural features described in words. The reader may get confused without a scheme and then have issues in following the mechanism clearly.

Response: Yes, “cluster”, “pyramid” and “pyramid center” define the same system: the clustered Trp tetrahedron by W233, W285, W337 and W94b. UVR8 Trp residues were classified into three groups (reference 20). This paper mainly focuses on one group: the Trp pyramid. Thus, the positions of other Trp residues were not shown in Fig. 1. For general readers, we have modified the sentence defining the three groups of Trp at line 73-79. We have included the residues

numbers and cited a previous paper. The figures in the reference 20 clearly show the location of all Trp residues and the definition of different Trp groups.

4) Page 4, line 98. Now the authors use the words "pyramid-cluster". They should stick to the same wording.

Response: We have changed “pyramid cluster” to “pyramid”.

5) Page 4, line 102. It is not clear why W233 would be an energy sink. It is apparently an electron donor. I would say that the energy sink (i.e. in terms of reaction energy) is the W285-W233 dimer.

Response: We have elucidated the excitation energy-transfer network in UVR8 in previous studies (reference 20). W233 has unusually red absorption/emission and long fluorescence lifetimes (~12 ns). This makes W233 an excellent excitation energy acceptor. All other Trp residues funnel energy to this site, increasing yield of charge separation. We have included citation of reference 20 here.

6) Page 4. Several places (this comment is extended to the full paper). It is important to use the correct symbols and naming for reaction intermediates. So, after the ET reaction W285 is a radical-anion and W233 is a radical-cation. So W^+ is not correct but it has to be $W^{+\cdot}$ and not W^+ . There are other cases of inconsistencies throughout the manuscript.

Response: This is a good point. We have changed all W^+ to $W^{+\cdot}$, $W233^+$ to $W233^{+\cdot}$ and $W285^-$ to $W285^{\cdot-}$ in the main text as well as in the supporting information to distinguish them from closed-shell cation/anion.

7) Page 5. The mechanism requires two sequential electron transfers $W285^- \rightarrow W233^+$ and then $R \rightarrow W285^-$. This has to be better clarified. In a sense a general scheme (may be in the Conclusion section) with the different parts of the proposed mechanism would greatly help the reader.

Response: This a good point. We have added the following sentence to the conclusion part “In summary, the proposed mechanism involves two sequential electron transfer steps: excited W233 to W285 and then $W285^{\cdot-}$ to $R286^+/R338^+$, followed by $W233^{+\cdot}$ deprotonation to yield $W233^{\cdot}/Arg^{\cdot}$ radical pair.”

8) Page 6, line 148-150. This sentence has to be rephrased or corrected to improve readability.

Response: This is a good point. We have rewritten the sentence as the following “The transient absorption decay of W285F (grey circles in Fig. 2e) agrees with its W^* decay dynamics (the red

curve) from the above fluorescence detection, excluding $W^{+\bullet}$ formation in the mutant. Conversely, the WT transient (grey squares in Fig. 2e) is clearly different from the W^* dynamics in the pyramid center (the dark yellow curve in Fig. 2e), suggesting $W^{+\bullet}$ contribution in the signal.”

9) Page 6, line 151. It seems that "one" should be two and "two" should be one.

Response: We think the current text is correct. As shown in supporting information equation 9, tryptophan cation radical population dynamics have 3 exponential components. The “two rises” are the 3.2-ps and 80-ps rise components. The “one decay” is the 3-ns decay due to $W233^{+\bullet}$ deprotonation.

10) Page 9, line 212. It is not clear how the two R286 and R338 radical molecular mechanics parameters have been generated.

Response: Only atomic charges were modified in the MD simulations. MSDFT was applied to calculate atomic charges of radicals ($W233^{+\bullet}/W285^{\bullet}$, $W233^{\bullet}/R286^{\bullet}$ and $W233^{\bullet}/R338^{\bullet}$) following the procedure described in reference 35. Briefly, gas phase MSDFT calculations were performed on residue pairs by PBE0 functional with HF correction for the off-diagonal Hamiltonian matrix element with 6-31+G(d) basis set.

We have included the following sentences in Methods section “For atomic charges of radical pairs, gas phase MSDFT calculations were performed on residue pairs by PBE0 functional 6-31+G(d) basis set was used for all QM calculations.”

11) Page 9, line 222. The large conformation changes need to be briefly described in the main text.

Response: Large conformations mainly mean disruption of interfacial hydrogen-bond networks, increased number of water molecules at the interface and partial unwinding of the dimer. To avoid confusion, we have deleted the citation of Supplementary Fig. 9d. We also added more discussions about conformational changes in this section.

12) Page 10. Conclusion section. Ss already mentioned above a general scheme of the entire mechanism should be given.

Response: Same as point number 7, we have added the following sentence to the conclusion part “In summary, the proposed mechanism involves two sequential electron transfer steps: excited $W233$ to $W285$ and then $W285^{\bullet}$ to $R286^{\bullet}/R338^{\bullet}$, followed by $W233^{+\bullet}$ deprotonation to yield $W233^{\bullet}/Arg^{\bullet}$ radical pair.” This should concisely summarize the proposed mechanism.

13) Page 15, line 355. CMAP has to be explained.

Response: According to reference 34, CMAP refers to inclusion of an energy correction map, to minimize the difference between the empirical and QM target maps. We have included a short explanation following the word “CMAP”.

14) Page 15, line 358-359. What "patch residues" mean? Is the only parametrization done that of the point charges? Which particular charges have been generated using the QM calculation? Please, explain (in more details in the Supporting information).

Response: “Patch residues” means the forcefield-modified residues (W^* , W^{+*} , W^* , W^* , R^*) that are separately defined in “patch” files for the MD simulation. Only atomic charges (treated as point charges) were modified in the MD simulations. Atomic charges of the modified residues were calculated using time dependent density functional theory (TDDFT) and multistate density functional theory (MSDFT). MSDFT was applied to constrain the charges in calculation of diradical states ($W233^{+*}/W285^{*-}$, $W233^*/R286^*$ and $W233^*/R338^*$). Atomic charges of $W233^*$ was calculated by TDDFT. We have included more explanations in the methods and SI.

15) Page 15, line 360. Have the use of TDDFT and MSDFT (with the functionals used in the paper) been benchmarked for excitation energy and charge transfer descriptions? Explain carefully for the general readership. Have TDDFT and MSDFT been used for excitation energies and for the description of the charge transfer state respectively? Explain better (very little details are given in the supporting information on these core issues).

Response: MSDFT was used to construct the diabatic states of electronic localized excitation and electron transfer. In addition, MSDFT was used to calculate electronic coupling matrix element for rate constants. TD-CAM-B3LYP together with MSDFT was used to obtain the locally excited $W233$, for the calculation of electronic coupling between $W233^*/W285$ and $W233^{+*}/W285^{*-}$. Without MSDFT, Trp excitation tends to delocalize among $W233$ and $W285$, and thus we were not able to model locally excited $W233$ only with TDDFT.

This details about this method were described in SI reference 12 (Ren, H. S. *et. al.*, J. Phys. Chem. Lett. 7, 2286-2293, 2016) and SI reference 13 (Gao, J. L. *et. al.*, J. Phys. Chem. Lett. 7, 5143-5149, 2016). In the papers, the computed transfer integrals using this method are consistent with experimental data and show the expected exponential attenuation with the donor–acceptor separation in a series of model systems.

To make clear, the following sentences were integrated into the QM/MM calculations part of the revised Methods section and SI.

“MSDFT was used to construct the diabatic states of electronic localized excitation and electron transfer. TD-CAM-B3LYP together with MSDFT was used to obtain the locally excited $W233$. In addition, MSDFT was used to calculate electronic coupling matrix element between two diabatic states for rate constants.”

16) Pag. 4 Supporting Information. The diabatisation process is not clear enough. a and b are residues or monomers? Please, explain exactly to which particular moieties the wave functions refer to.

Response: Thank the reviewer for pointing out. a and b are different residues. To clarify, we have changed “monomer b” to “residue b” in the SI.

REVIEWERS' COMMENTS

Reviewer #1 (Remarks to the Author):

I am happy with the authors' responses and am now happy to recommend publication

Reviewer #2 (Remarks to the Author):

The authors have clarified all my questions and concerns. I recommend the paper for publication.

Reviewer #3 (Remarks to the Author):

I have carefully examined the reply to my own criticism as well as the reply to the criticism to the other reviewers. I am convinced that the authors provide both convince and exhaustive replies. In particular, I have appreciated their willingness to explain in details how the mechanism proposed compare with the previously reported mechanism and why these present shortcomings that the present research has tried to overcome. I have also appreciated the willingness of the authors to detail the different aspects of their computational methodology which, in principle, could now be replicated (although with a rather big effort). In conclusion, also on the basis of a reevaluation of the manuscript, I have no further points to raise and I recommend acceptance.

Reply to reviewers

Reviewer #1:

No questions anymore; accept for publication!

Reviewer #2:

No questions anymore; accept for publication!

Reply to Reviewer #3:

No questions anymore; accept for publication!